# Genome-Scale Identification of Wild Soybean Serine/Arginine-Rich Protein Family Genes and Their Responses to Abiotic Stresses

**DOI:** 10.3390/ijms252011175

**Published:** 2024-10-17

**Authors:** Yanping Wang, Xiaomei Wang, Rui Zhang, Tong Chen, Jialei Xiao, Qiang Li, Xiaodong Ding, Xiaohuan Sun

**Affiliations:** 1Mudanjiang Branch of Heilongjiang Academy of Agricultural Sciences, Mudanjiang 157000, China; wyanping12@gmail.com (Y.W.); wangxiaomei278@gmail.com (X.W.); 2Key Laboratory of Agricultural Biological Functional Genes, College of Life Science, Northeast Agricultural University, Harbin 150030, China; rui952569@gmail.com (R.Z.); tchen1237@gmail.com (T.C.); xiaojialei1978@neau.edu.cn (J.X.); lstrong@neau.edu.cn (Q.L.);

**Keywords:** wild soybean, SR protein, alternative splicing, abiotic stress

## Abstract

Serine/arginine-rich (SR) proteins mostly function as splicing factors for pre-mRNA splicing in spliceosomes and play critical roles in plant development and adaptation to environments. However, detailed study about SR proteins in legume plants is still lacking. In this report, we performed a genome-wide investigation of SR protein genes in wild soybean (*Glycine soja*) and identified a total of 31 *GsSR* genes from the wild soybean genome. The analyses of chromosome location and synteny show that the *GsSR*s are unevenly distributed on 15 chromosomes and are mainly under the purifying selection. The GsSR proteins can be phylogenetically classified into six sub-families and are conserved in evolution. Prediction of protein phosphorylation sites indicates that GsSR proteins are highly phosphorylated proteins. The protein–protein interaction network implies that there exist numerous interactions between GsSR proteins. We experimentally confirmed their physical interactions with the representative SR proteins of spliceosome-associated components such as U1-70K or U2AF35 by yeast two-hybrid assays. In addition, we identified various stress-/hormone-responsive *cis*-acting elements in the promoter regions of these *GsSR* genes and verified their expression patterns by RT-qPCR analyses. The results show most *GsSR* genes are highly expressed in root and stem tissues and are responsive to salt and alkali stresses. Splicing analysis showed that the splicing patterns of *GsSR*s were in a tissue- and stress-dependent manner. Overall, these results will help us to further investigate the biological functions of leguminous plant SR proteins and shed new light on uncovering the regulatory mechanisms of plant SR proteins in growth, development, and stress responses.

## 1. Introduction

It is estimated that 80% of plant nuclear genes are interrupted by non-coding introns. To generate mature functional mRNA from intron-containing precursor mRNA (pre-mRNA), the co-transcriptional process of pre-mRNA splicing, including constitutive splicing (CS) or alternative splicing (AS), is indispensable [1]. The splicing process is performed by the spliceosome, a large multi-component megadalton complex, which consists of five small nuclear ribonucleoproteins (snRNPs) and numerous spliceosome-associated auxiliary proteins [2]. Among these auxiliary proteins, the serine/arginine-rich proteins (SR proteins) play vital roles in the most crucial and early steps of spliceosome assembly, thereby contributing to the regulation of the transcriptome and proteome diversity of organisms [3].

The SR proteins are highly conserved and are widely present in eukaryotes such as metazoans, plants, and fungi [4]. SR proteins are characterized by one or two RNA recognition motifs (RRMs, PF00076) in their N-terminal regions and an arginine/serine (RS) region with at least 50 amino acids characterized by consecutive RS or SR repeats in their C-terminal regions. The RRM domains can recognize and bind to a variety of pre-mRNA *cis*-regulatory elements, and the RS region is intrinsically disordered, which is required for protein–protein interactions. The previous interaction studies have revealed a complex network of direct interactions among SR proteins and the other proteins (e.g., SnRNPs, protein kinases, and RNA polymerase II) [1,5].

The first plant SR protein was identified in 1990s from Arabidopsis [6]. To date, many SR family proteins have been identified in a subset of plant species, such as rice (*Oryza stavita*) [7], cotton (*Gossypium spp.*) [8], *Brassica napus* [9], longan (*Dimocarpus longan*) [10], wheat (*Triticum aestivum*), *Brachypodium distachyon* [11], and so on. Compared with the animal kingdom, plant genomes harbor more SR protein-encoding genes and greater diversity of SR proteins, which probably evolved for plant-specific functions [12]. According to their domain organization, the plant SR protein family can be divided into six sub-families: SR, RSZ, SC, SCL, RS2Z, and RS [13]. Among them, the SCL, RS2Z, and RS sub-families are plant-specific, whereas the other three sub-families (SR, RSZ, and SC) have orthologs in animals. The SR proteins in SR and RS sub-families contain two RRM and one RS domain. The SR sub-family possesses a conserved SWQDLKD motif in the second RRM. The members of the other four sub-families (SC, SCL, RSZ, and RS2Z) contain a single RRM domain and RS domain. The SCL members contain an extra N-terminal charged extension, and RSZ members contain one zinc knuckle separating RRM from RS domain, and RS2Z sub-family proteins contain two zinc knuckles and an extra SP-rich region.

Previous studies have shown that the SR proteins widely participate in plant tolerance to abiotic stresses. In Arabidopsis, the expression of the *AtSR34b* gene could be upregulated by cadmium (Cd^2+^) stress, and another study indicates that under Cd^2+^ stress, *atsr34b* mutant plants had a splicing defect in the iron-regulated transporter 1 (*IRT1*) gene, thus reducing the iron-regulated transporter 1 (*IRT1*) mRNA accumulation and causing plant sensitivity to heavy metal stress [14]. Moreover, plant-specific AtSCL30a negatively regulates ABA signaling to control seed traits and stress responses during germination in Arabidopsis [15]. Overexpression of *OsSCL30* diminished plant resistance to cold, drought, and salt stresses in transgenic Arabidopsis and rice and resulted in a large accumulation of reactive oxygen species (ROS) [16]. However, the study of SR proteins’ response to abiotic stresses is still limited in leguminous crops.

Cultivated soybean (*Glycine max*) is the most important legume crop species worldwide, particularly for its rich seed protein and oil contents for human and animal consumption [17,18]. Wild soybean (*Glycine soja*) is the closest species of cultivated soybean, possessing a much greater adaptability and more genetic variations to environmental stresses such as drought, alkali, salt, and barren soil [19,20]. It has therefore been suggested that wild soybeans are potential genetic germplasm resources for breeding and improvement of soybean varieties. As we know, SR proteins play critical roles in gene splicing and plant responses to environmental stresses [3]; however, the study of the SR protein family and its role in response to environmental stresses in legumes is still elusive. In this study, we conducted a comprehensive analysis of wild soybean SR protein genes at the genome-wide scale, which lays a foundation for further functional elucidation and utilization of SR genes in this family.

## 2. Results

### 2.1. Identification and Characterization of SR Protein Genes in Wild Soybean

A total of 31 candidate coding sequences for GsSR protein family were obtained. Analysis of these GsSR protein gene family members revealed that these 31 *GsSR*s are distributed across 15 out of total 20 wild soybean chromosomes (chromosome 1-10, 14, 16-17, 19-20). The coding sequence (CDS) length of the *GsSR* genes ranges from 336 to 1131 bp, and the protein length of the GsSR proteins varies from 111 to 376 amino acids. The relative molecular weights of the GsSR proteins are between 12.96 kDa and 43.27 kDa, and, notably, all of the GsSR proteins except GsSR9 (pI = 6.9) are alkaline proteins with the isoelectric point (pI) ranging from 9.19 to 11.42 (Table 1). 

According to the subcellular localization prediction, GsSR2 and GsSR21 are localized in the Golgi apparatus, GsSR1 and GsSR24 are localized in the plastid, GsSR23 is localized in the extracellular, and the remaining 26 members are all localized in the nucleus. These results indicate that most SR proteins from wild soybean are nuclear proteins, which is consistent with their functions as splicing factors.

### 2.2. Phylogeny of Wild Soybean SR Protein Genes

To analyze the evolutionary relationships between the SR protein gene families in wild soybean and the model plant species (Arabidopsis and rice), we constructed a comparative phylogenetic tree by using the maximum likelihood (ML) method from multiple sequence alignments of 31 GsSR, 18 AtSR, and 22 OsSR protein sequences (Figure 1). Similar to Arabidopsis and rice SR protein families, the wild soybean SR proteins can be divided into six sub-families: RS, SR, RSZ, RS2Z, SC, and SCL. Among these sub-families, the SCL, RS, and RS2Z sub-families are plant-specific, and they have the most members, i.e., eight, six, and six, respectively, in wild soybean. In contrast, the SC sub-family has the least members, only two in wild soybean (GsSR10 and GsSR15).

### 2.3. Analysis of Chromosome Location, Duplications, and Synteny of SR Protein Genes in Wild Soybean

To examine their chromosomal distributions, we mapped all the 31 *GsSR* genes onto 15 chromosomes based on the physical positions (Figure 2A, Table 1). The 31 *GsSR* genes are unevenly distributed on 15 chromosomes. Chromosome (Chr) 3 contains four *GsSR* genes. Chr 2, 7, 8, 14, 17, and 19 contain three *GsSR* genes. Chr 20 contain two *GsSR* genes, and Chr 1, 4, 5, 6, 9,10, and 16 only have one *GsSR,* respectively.

It is reported that gene duplication events play crucial roles in amplification of gene family members in the genome [21]. In order to clarify the expansion mechanism of the GsSR protein gene family, we therefore analyzed tandem duplicated (TD) and segmental duplication (SD) events of GsSR protein gene members. For TD events, four *GsSR* genes were clustered into two TD clusters within the same chromosome regions (*GsSR6/7* and *GsSR27/28*). Although *GsSR15* and *GsSR16* formed another tightly linked *GsSR* cluster, they could not be considered a TD cluster because they belong to different sub-families with poor sequence similarity. For SD events, a total of 16 gene pairs among different chromosomes were implicated in SD events, but only eight *GsSR* genes were found to be located outside any of the duplicated blocks (Figure 2B), suggesting that SD takes a major part in expansion of the SR protein gene family in wild soybean. In addition, we calculated the *Ka*/*Ks* ratios of all *GsSR* SD pairs. The results showed that most of the *GsSR* SD pairs underwent purifying selection, except the SD pair *GsSR2/GsSR18,* whose *Ka*/*Ks* ratio is larger than 0.9 (Appendix A), which is indicative of positive selection acting on the genes [4].

To further understand the gene replication, the synteny relationships among SR protein genes from wild soybean, Arabidopsis, and rice were analyzed. The results showed that out of 31 *GsSRs*, 19 *GsSRs* had 30 pairs of collinearity with Arabidopsis, and among these 19 *GsSRs*, 9 *GsSRs* had single pairs, and another 9 *GsSRs* had two pairs, and the remaining *GsSR* had three pairs of SR collinearity between wild soybean and Arabidopsis, while for the collinearity between wild soybean and rice, only four *GsSRs* had eight pairs of collinearity with rice. Among these four *GsSRs*, one *GsSR* has single or three pairs, respectively, and two *GsSRs* have two pairs (Figure 2C and Appendix A). Our results reveal that the dicot wild soybean has more homologous SR protein genes with dicot species (Arabidopsis) than monocot species (rice).

### 2.4. Analysis of Structure and Motif of GsSR Genes and Proteins

In eukaryotes, the gene structure conservation may be a record of key evolutionary events [22]. In order to investigate the exon–intron structural evolution of SR protein gene paralogs in wild soybean, we analyzed the structures of all GsSR protein genes. As shown in Figure 3B, the structures of 31 *GsSR* genes in the six sub-families were displayed. On average, the *GsSR* genes have 6.6 introns, but the intron numbers for each member differ widely, ranging from 1 (*GsSR9*) to 14 (*GsSR8* and *GsSR29*). Generally, the exon/intron numbers are relatively conserved within a sub-family. For example, all *GsSR* genes in SC sub-family have 9 introns, and all *GsSR* genes in SCL and SR sub-families have 5 to 7 introns except *GsSR9* (1 intron), and all *GsSR* genes in RS sub-family have 4 to 5 introns except *GsSR13* (8 introns). Comparatively, the numbers of introns in the members of SR (3 to 14 introns) and RSZ (4 to 9 introns) sub-families vary dramatically.

Furthermore, we analyzed the conserved motifs of GsSR proteins. As shown in Figure 3C, all of the identified GsSR proteins possess one or two RRM motifs at their N termini and are highly conserved in the types and distribution of conserved domains in the same sub-family. For instance, the members of SR and RS sub-families have a second RRM domain, and the members of RSZ and RS2Z have one or two ZnF_C2HC motifs, and the members of SCL sub-family have an extra-low-complexity region at their N termini. Nevertheless, some members of certain sub-families might acquire new domains or lost conserved domains during duplication events. For example, the transmembrane domains were identified in members of the RSZ (GsSR21) or SCL (GsSR2) sub-family, the internal repeat 1 (IR1) domain was found in members of the RS sub-family (GsSR23) and SCL sub-family (GsSR22), two members of the SR sub-family (GsSR1 and GsSR25) lost their second RRM domain, and two members of RS2Z sub-family (GsSR9 and GsSR16) lost two ZnF_C2HC motif domains. 

### 2.5. Analysis of Cis-Acting Elements in GsSR Gene Promoter Regions

As key factors in the early recognition of splice sites, plant SR proteins and their gene expressions are usually regulated by various developmental and environmental cues [3,23], so we tried to identify the stress/hormone-associated *cis*-acting elements on the promoter regions of these *GsSR* genes. As shown in Figure 4 and Appendix A, many stress/hormone-responsive *cis*-acting elements were found in their promoter regions. The responsive stresses include anaerobic stress, drought, wound, low temperature, and mixed stresses, and the responsive phytohormones include auxin, MeJA, gibberellin, salicylic, and abscisic acid. Among all the stress/hormone-responsive *cis*-acting elements, the *cis*-acting element with the largest number is ABRE, an abscisic acid-responsive *cis*-acting element, followed by ARE, an anaerobic-responsive *cis*-element. Of all the *GsSR* members, the promoter region of *GsSR13* (belongs to RS sub-family) has 18 stress/hormone-responsive *cis*-acting elements, more than those in any other *GsSR* gene promoters. In contrast, the promoter of *GsSR19* (also belongs to the RS sub-family) has only one stress/hormone-responsive *cis*-acting element.

### 2.6. Prediction of Phosphorylation Sites in GsSR Proteins

Most of SR proteins are extensively phosphorylated, and phosphorylation modification plays a considerable role in controlling SR protein activities [24]. Hence, we predicted the putative phosphorylation sites in GsSR proteins. A total of 1699 nonredundant phosphorylation sites (pSites) were identified from 31 GsSR proteins, and the numbers of predicted GsSR pSites range from 31 (GsSR26) to 74 (GsSR31) with an average of 58 phosphosites (Table 1), so it is postulated that the SR proteins might be highly phosphorylated for their splicing activities. The statistical data showed that among all pSites, most (65%) are phosphorylated on serine (Ser) sites, followed by tyrosine (Tyr) (24%) and threonine (Thr) (11%) residues (Figure 5A).

To predict the potential upstream protein kinases of GsSR proteins, we enriched the conserved motifs around the phosphosites. The flanking sequences (seven amino acids upstream and downstream) of putative pSer, pThr, and pTyr, respectively, were pre-aligned, and then the overrepresented motifs were extracted. As shown in Figure 5B and Appendix A, ten motifs ([SXSXpSP], [RpSXSP], [DXXXpSP], [RSXpSP], [RXRpSRS], [RXXpSXSP], [pSP], [RpSXSXXXS], [pSXSP], and [pSXXL]) were extracted from the flanking sequences with pSer, and two motifs ([SXXpYG] and [pYXXXXXY]) were extracted from the flanking sequences with pTyr, whereas no motif was enriched from the flanking sequences with pThr.

### 2.7. Interaction Analysis of GsSR Proteins

To regulate constitutive splicing (CS) or alternative splicing (AS) of the target pre-mRNAs, the splicing machineries (spliceosomes) can be assembled through physical interactions of SR proteins and other proteins [12]. To further understand the functional mechanisms of GsSR proteins and their interactions among different GsSR proteins and other proteins, we generated an interaction network of GsSR proteins. As shown in Appendix A and Appendix A, 118 protein pairs were predicted with high confidence (score > 0.700) among 21 GsSR proteins, and the average edge number per GsSR protein node is 11.2. Further analysis indicates that some GsSR protein nodes have much more connected edges than the average edge number of GsSR proteins. For example, GsSR15 (20 edges), GsSR4 and GsSR20 (18 edges each), and GsSR14 and GsSR24 (17 edges each) are observed to have the highest connectivity, implying these SR proteins may actively participate in spliceosome assembly.

The interactions between SR proteins and small nuclear ribonucleoproteins (snRNPs) are important for splicing activity [5]. To elucidate the relationships between GsSR proteins and snRNPs (i.e., GsU1-70K or GsU2AF^35^ proteins), we examined the physical interactions between six GsSR proteins and two typical snRNP proteins (GsU1-70K or GsU2AF^35^) by using a Y2H assay. Six representative GsSR proteins were picked from each sub-family of SR proteins (Appendix A). As shown in Figure 6, the yeast cells co-transformed with AD-GsSR8/12/15/26 and BD-GsU1-70K or BD-GsU2AF^35^ or the positive and negative control plasmids exhibited equal growth on the SD-L-T medium. However, the cell harboring above plasmids showed different growth on SD-L-T-H medium, suggesting that these SR proteins have a different interacting relationship with snRNPs. GsSR members (GsSR8, GsSR12, GsSR15, and GsSR26) may interact with both GsU1-70K and GsU2AF^35^. The yeast cells co-transformed with AD-GsSR22 and BD-GsU1-70K but not BD-GsU2AF^35^ were able to grow on SD-L-T-H medium, indicating GsSR22 can specifically interact with GsU1-70K but not GsU2AF^35^. The yeast cells co-transformed with AD-GsSR16 and BD-GsU1-70K or BD-GsU2AF^35^ could not grow on SD-L-T-H medium, indicating that GsSR16 could not interact with either GsU1-70K nor GsU2AF^35^. Taken together, these GsSR proteins may play distinct roles in spliceosome assembly and pre-mRNA splicing.

### 2.8. Expression Patterns of GsSR Genes in Different Tissues Under Abiotic Stresses

To expand our knowledge about the biological function of SR protein genes, we analyzed the spatial expression patterns of *GsSR* genes in different tissues of wild soybean by RT-qPCR analyses. As shown in Figure 7, the selected *GsSR* genes showed significant variation in root, stem, leaf, and pod tissues. Overall, most *GsSR* genes were highly expressed in roots followed by leaves, whereas the expression levels of *GsSR* genes in pods were relatively the lowest. Moreover, some genes, such as *GsSR24*, had high expression levels in most tissues, whereas the other genes, such as *GsSR5* and *GsSR14*, had relatively low expression levels in most tissues.

In addition to the expression patterns in different tissues, the expression profiles of 20 selected *GsSR* genes under salt stress (150 mM NaCl) and 10 selected *GsSR* genes under alkali stress (150 mM NaHCO_3_) in root tissues were analyzed by RT-qPCR (Figure 8). The results showed that the expression patterns of different *GsSR* genes show significant differences under both stresses. Almost all *GsSR* genes were significantly upregulated or downregulated by NaCl and NaHCO_3_ treatments at certain time points, indicating that these *GsSR* genes might be involved in response to salt and alkali stresses in wild soybean. Intriguingly, the expression patterns of the same gene under salt and alkali stresses show a high degree of consistency. For instance, the expression levels of *GsSR23/24/28* reached their peaks at 3 h after salt or alkali treatment, and the expression level of *GsSR7* reached its peaks at 6 h after both treatments, and the expression of *GsSR4* was suppressed after subject to both stresses. Overall, most of *GsSR* genes are actively responsive to different abiotic stresses in wild soybean.

### 2.9. Alternative Splicing of GsSR Pre-mRNAs

SR proteins play critical roles in both CS and AS of pre-mRNAs, and, intriguingly, most *SR* pre-mRNAs themselves in plants also undergo alternative splicing (AS). The AS of *SR* pre-mRNAs is regulated by various environmental stresses [7,25]. To understand the AS patterns of *SR* genes in wild soybean, we identified a total of 93 transcripts from 31 *GsSR* genes by searching the wildsoydb database (Appendix A). We found that a total of 23 *GsSR* genes undergo AS, and on average each gene produces 2 or more transcripts, respectively. Particularly, we detected 9 transcripts for *GsSR10* gene and 7 transcripts for *GsSR30* genes (Appendix A). Furthermore, we also found that all the *GsSR* genes from the six sub-families have AS events, and the average transcript numbers of each sub-family are somewhat similar, varying from 2.0 to 3.5 (Appendix A).

To further analyze and validate the splice events of *GsSR* genes, we treated the wild soybean seedlings with salt and alkali stresses and extracted total RNA from leaves and roots for RT-PCR analyses by using the primers (Appendix A) that were specific to the eight selected *GsSR* genes (*GsSR3*, *6*, *11*, *14*, *15*, *17*, *29*, and *30*). The RT-PCR analyses showed that all these eight *GsSR* genes exhibited alternative splicing with two to four distinct transcripts in different tissues and under different stresses (Figure 9), suggesting that the wild soybean SR genes are actively and alternatively spliced in a stress- and tissue-dependent manner.

## 3. Discussion

Global analyses of splicing of precursor messenger RNAs (pre-mRNAs) have revealed that alternative splicing (AS) is highly pervasive in plants. Despite the widespread occurrence of AS in plants, the mechanisms that control splicing and the roles of splice variants generated from a gene are poorly understood. Plants have many more SR proteins compared to animals, including several plant-specific subfamilies. Pre-mRNAs of plant SR proteins are extensively alternatively spliced to increase the transcript complexity by about six-fold. AS of SR pre-mRNAs is altered by various stresses, raising the possibility of rapid reprogramming of the whole transcriptome by external signals through regulation of the splicing of these master regulators of splicing [26]. In the present study, we performed a genome-wide analysis of SR protein family genes in wild soybean by using multiple bioinformatic tools. Our study provides a comprehensive understanding of the SR protein family in plant species and facilitates further investigations on the gene and protein structures and the classification and evolution of plant SR proteins. Moreover, the expression patterns, protein interactions, and alternative splicing analyses provide us with insight into the biological functions and mechanisms of plant SR proteins in response to abiotic stresses. Here, we just compared the SR genes and proteins with those from monocot and dicot model plants (Arabidopsis and rice). Furthermore, it will be intriguing to compare the SR genes in the cultivated soybean and identify the promising genes for soybean breeding. 

### 3.1. The GsSR Protein Family Expansion Depends on Various Duplication Mechanisms

A total of 31 SR protein encoding genes were identified from the wild soybean genome in this study, accounting for 0.056% of annotated wild soybean protein-coding genes. The number of SR proteins in wild soybean is larger than that of Arabidopsis (18 SR proteins) and rice (22 SR proteins). These GsSR proteins are classified into six sub-families, and the proportion of three plant-specific SR sub-family (RS, RS2Z, and SCL) members in wild soybean was up to 64.5% (20/31), which is also higher than those of Arabidopsis (55.56%, 10/18) and rice (54.5%, 12/22), indicating these three plant-specific sub-families are greatly expanded in wild soybean. The large number of SR protein genes in flowering plants can be attributed to whole genome duplication (WGD) events and due to two WGD events. Until now, it has been known that soybean has the largest number of *SR* genes in almost all studied flowering plants [4]. Furthermore, the duplication mechanisms underlying GsSR protein family expansion also include TD and SD events. In our study, we identified two TD clusters and 16 SD pairs from all *GsSR* genes, and only eight *GsSR* genes were outside any of the duplicated blocks. These results suggest that SD took a great part in GsSR protein family expansion. In addition, the *Ka*/*Ks* ratios analysis shows that the SD pairs of *GsSR* genes underwent strong purifying selection. The whole-genome sequencing of 185 diverse wild soybean accessions collected from three major agro-ecological zones in China exhibited clear agro-ecological zone-based population structure and multiple environmental factors [27].

### 3.2. GsSR Proteins Are Conserved on Gene Structure and Protein Motif Levels

In general, the exon–intron structures and protein motifs of different SR members are conserved within the same sub-families. However, losing/gaining conservative domains in certain SR protein members may lead to defunctionalization or neofunctionalization. Here, our study showed all GsSR proteins have one or two highly conserved N-terminal RRM domains and a C-terminal divergent RS domain. The role of the RRM domain is related to RNA binding, albeit the RNA binding motifs are usually variable within RRM regions. Therefore, confirmations of the RNA binding residues by site-directed mutagenesis experiments are required in future study. The RS domains are mostly required for essential SR protein function and intrinsically disordered feature [4], which is shown as the “low complexity regions (LCR)” in the C-terminal of each GsSR protein member (Figure 3C). The LCRs, also named IDRs (intrinsically disordered regions), and RRM domains provide a critical foundation for forming liquid–liquid phase separation (LLPS) within cells [28]. Hence, LLPS appears to be a common feature in SR protein family. Fargason et al. reported that a human SR protein SRSF1(SF2/ASF) underwent LLPS in physiological buffer [29]. Our recent study also suggested that SRRM1L protein, an Arabidopsis SR-like protein with RS domain, undergoes in vivo and in vitro LLPS, which plays an important role in the accumulation of SRRM1L in nuclear speckles in response to salt stress [30].

### 3.3. The Expression and Splicing Patterns of GsSR Protein Genes Are Tissue- and Stress-Dependent

We identified dozens of stress-/hormone-responsive *cis*-acting elements from the *GsSR* gene promoters, implying that SR proteins may be widely involved in modulating plant growth, development, and various responses to stresses. Further expression analyses showed that most *GsSR* genes are mainly expressed in roots and stems and were able to respond to salt and alkali stresses. The tissue-specific and stress-induced expression patterns of *GsSR* genes indicate these proteins may actively participate in regulating plant adaptation to environments, consistent with the *SR* genes in other plants [26].

Moreover, our study identified several *GsSR* genes (such as *GsSR4*, *7*, *23*, *24,* and *28*) displaying similar expression patterns under both salt and alkali stresses, indicating that plants may have common pathways in response to distinct stresses, and these *GsSR* genes may be functionally redundant in wild soybean. The same SR protein has been reported to participate in various stresses in other species. In rice, the expressions of *OsSC25* and *OsRSZ21a* genes were regulated by both drought and cold stresses, while the expressions of *OsSC32* and *OsRSZ23* were modulated by both salt and temperature stress [7]. In cassava (*Manihot esculenta*), the *MeSCL30A* transcript was responsive to both drought and salt stresses [31].

Based on the analyses of different expression patterns of *GsSR* genes, we also determined their splicing patterns in the present study. A total of 93 transcripts were identified from 31 *GsSR* genes, indicating that these SR proteins not only modulate the splicing of downstream target genes as key splicing factors, but also these SR proteins themselves are also universally regulated by AS. The AS of both *SR* pre-mRNAs and their downstream targets contribute greatly to the complexity of transcriptome and proteome in cells. The AS of SR pre-mRNAs may be auto-regulated or regulated by other SR proteins. In Arabidopsis, SCL33 mediates the auto-regulation of alternative splicing of its third intron by recognizing the intronic *cis*-element GAAG, and another SR protein SCL30a also participates in the regulation of this AS event [32]. In this work, our RT-PCR results indicate that the splicing patterns of GsSR protein genes were specific to different tissue/stress combinations, indicating that the GsSR proteins can modulate plant responses to abiotic stresses at both the gene expression and transcript/AS levels.

### 3.4. Phosphorylation Modification on Post-Translational Level Is a Crucial Regulatory Means for GsSR Proteins 

SR proteins are highly phosphorylated splicing regulators that function in pre-mRNA splicing to modulate the developmental processes and adaption to environmental stresses in plants. Post-translational modifications, primarily phosphorylation, determine the biological functions of SR proteins and their splicing activities [33,34]. In the present study, we identified an average of 58 putative pSites on each GsSR protein member. However, the upstream protein kinases of SR proteins are still elusive. Our early study in rice identified the interactions of several SR proteins (OsSCL30a, OsSCL30b, OsRSZp23, and Os06g50890) with a rice LAMMER kinase by using a high-throughput yeast two-hybrid system [35]. Multiple studies have reported that LAMMER kinases are involved in regulation of pre-mRNA processing and splicing [36,37]. Our recent phosphoproteomics study on soybean under alkali stress identified an interaction combination of GmPRP4KA kinase and GmSC35, and the physical interaction was further verified by yeast two-hybrid and split-LUC complementation assays [38], implying that PRP4KA kinase may participate in the regulation of SC35 protein by direct phosphorylation. Moreover, our other phosphoproteomics study on Arabidopsis also found five SR/SR-like proteins (SRRM1L, RS40, RS2Z32, RS2Z33, SCL30A and SR45) might be phosphorylated in a SnRK1 kinase-dependent pathway, implying that SnRK1 kinase also actively participates in the regulation of SR proteins [39]. In this study, by analyzing the flanking sequences of putative pSites, we identified 10 conserved phosphor motifs. Among them, the motif [pSP] was reported to be associated with GSK-3, cyclin-dependent kinase (CDK), and MAPK kinases [40,41], indicating that these protein kinase families may also be associated with the phosphorylation of SR proteins. Considering the importance of the phosphorylation modification to the regulation of SR protein function, it would be in our interest to identify more unknown upstream protein kinases of SR proteins and determine the underlying regulatory mechanisms of SR protein phosphorylation in developmental and abiotic stress responses in plants.

## 4. Materials and Methods

### 4.1. Identification of SR Genes in Glycine soja Genome 

The wild soybean genome assembly (W05) was downloaded from the wildsoydb database (http://www.wildsoydb.org/Gsoja_W05/ (accessed on 3 February 2023)) [42]. The SR protein gene sequences of Arabidopsis and rice in this study were downloaded from the EnsemblPlants database (http://plants.ensembl.org/index.html (accessed on 12 April 2024)). Both Basic Local Alignment Search Tool (BLAST) ((https://ftp.ncbi.nlm.nih.gov/blast/executables/blast+/2.7.1/ (accessed on 21 March 2024)) and Hidden Markov model (HMM) searches [43] were performed to detect all putative SR protein genes in the wild soybean genome. For local BLAST search analysis, the protein sequences of all 18 SR protein gene sequences from Arabidopsis were used as queries to search one by one against the wildsoydb database (E value was set to 1 × 10^−10^). For the HMM search, 18 AtSR and 22 OsSR protein sequences were selected to build an HMM profile and were then used to identify GsSR protein genes using the HMMER3.0 program (E value was set to 1 × 10^−5^). Both BLAST and HMM search results were combined to remove redundancy.

### 4.2. Subcellular Localization Prediction of GsSR Proteins

The subcellular localization of GsSR proteins was predicted by using Plant-mSubP online program [44], with “PseAACNCCDipep” as the selected prediction module.

### 4.3. Phylogenetic Relationship of GsSR Proteins 

The maximum likelihood (ML) phylogenetic tree was constructed by using TBtools software v2.096, which incorporates Muscle, trimAI, and IQ-tree programs for the multiple sequence alignment and tree construction. The phylogenetic tree was visualized by iTOL6 online program (https://itol.embl.de/ (accessed on 5 May 2024)) [45].

### 4.4. Analyses of Chromosome Location, Synteny, and Gene Duplication 

The locations of GsSR protein genes were obtained based on genome annotation, and the chromosomal distributions of GsSR protein genes were visualized using “Gene Location Visualize from GTF/GFF” function of TBtools v2.096 program [46]. The gene names from GsSR1 to GsSR31 were assigned on the basis of their positions from the top to the bottom of wild soybean chromosomes.

The tandem duplicated (TD) genes were considered an array of two or more genes tightly linked on the same chromosome within a 200 kb region. Then, the segmental duplication (SD) events of *GsSRs* and synteny relationships among wild soybean, Arabidopsis, and rice were analyzed by the MCScanX program [47], and the results were visualized using the “Advanced Circos” or “Multi Synteny Plot” functions of the TBtools v2.096 program.

### 4.5. Analyses of Gene Structure, Protein Motif, and Cis-Acting Elements Distribution

The structure information (intron–exon) of *GsSR* genes was extracted from the GFF/GTF files, the conserved motifs of GsSR proteins were analyzed using SMART (http://smart.embl-heidelberg.de/ (accessed on 7 June 2024)) [48], and the results were visualized using the “Gene Structure View (Advanced )” or “Simple Biosequence Viewer” functions of the TBtools v2.096 program.

The 2000 bp genomic DNA sequences upstream of the start codon of all *GsSR* genes were extracted by using the “Fasta Extract or Filter (Quick)” function of the TBtools v2.096 program, and then submitted to the PlantCARE website (http://bioinformatics.psb.ugent.be/webtools/plantcare/html/ (accessed on 10 June 2024)) for predicting the *cis*-acting elements. The result file was processed by using the “Simple PlantCARE Classify” function and the graphical figure was generated using the “Basic Biosequence View” function of the TBtools v2.096 program.

### 4.6. Prediction of Phosphorylation Sites of GsSR Proteins

The phosphorylation sites of GsSR proteins were predicted by using the NetPhos v3.1 service (https://services.healthtech.dtu.dk/service.php?NetPhos-3.1 (accessed on 10 April 2024)) [49]. The significantly enriched phosphorylation motifs around the putative phosphorylated amino acids were extracted by using the MoMo v5.5.5 online service (https://meme-suite.org/meme/doc/momo.html?man_type=web (accessed on 12 May 2024)) with the motif-x algorithm. The background dataset was based on soybean proteins, and the significance was set at 1 × 10^−6^ [50].

### 4.7. Prediction and Confirmation of Protein–Protein Interactions

The interactions among different GsSR protein members were predicted using the STRING online database (http://string-db.org/ (accessed on 13 June 2024)) [51] and visualized with the Cytoscape software (v3.9.1) [52].

The yeast two-hybrid (Y2H) was performed as described previously [38]. Briefly, the candidate *GsSR* protein genes and *GsU1-70K* or *GsU2AF^35^* genes were inserted into pGADT7 and pGBKT7 vectors, respectively. The obtained constructs were co-transformed into a Y2HGold yeast strain to determine the physical interactions of GsSR proteins with GsU1-70K or GsU2AF^35^. The vectors of pGADT7-GsSnRK1α and pGBKT7-GsSnRK1β were used as a positive control. The primer pairs for amplifying *GsSR* genes are listed in Appendix A.

### 4.8. Plant Growth and Stress Treatment

The seedlings of wild soybean line G07256 (from Jilin Academy of Agricultural Sciences, Changchun, China) were cultivated in a mixture of vermiculite, peat moss, and perlite (1:1:1) in a greenhouse under a 16 h/8 h (light/dark) photoperiod cycle at 25 °C with 60% relative humidity. For analysis of spatial expression patterns of the *GsSR* genes, the roots, leaves, stems, and young pods were collected from the 2-month-old plants. For stress treatments, the 3-week-old seedlings were transferred to a hydroponic system and were subjected to 150 mM NaCl or 150 mM NaHCO_3_ treatment for 0, 3, 6, and 12 h, respectively. There were at least three plants for each treatment with three biological repeats. All the samples were collected and frozen immediately in liquid nitrogen and stored at − 80 °C until total RNA extraction.

### 4.9. RNA Extraction, RT-PCR and RT-qPCR Analyses

Total RNA samples from wild soybean tissues were extracted using the EasyPure^®^ Plant RNA Kit (Transgen, Beijing, China), and cDNA was synthesized from 1.0 μg of total RNA using the Transcript^®^ALL in One First Strand cDNA Synthesis SuperMix kit (Transgen, Beijing, China), according to the manufacturer’s instruction. The RT-qPCR reactions were performed using 2× SYBR Green qPCR MasterMix (Toyobo, Osaka, Japan). The *GsGAPDH* gene was selected as an internal control. The experimental data were analyzed using the 2^−ΔΔCt^ method. Primer pairs of GsSR protein genes used for RT-qPCR validation are listed in Appendix A. The RT-PCR reactions were performed using 2× EasyTaq PCR SuperMix (TransGen, Beijing, China) with 5× diluted cDNA as the template. The PCR products were visualized in 1.0% agarose. Primer pairs of *GsSRs* used for RT-PCR validation are listed in Appendix A.

## 5. Conclusions

In this study, a total of 31 SR protein genes in wild soybean were identified and can be classified into six sub-families. These *GsSRs* are unevenly distributed on the chromosomes and are mainly under the purifying selection. The gene structure and protein motifs of the GsSR protein genes are conserved, and a large amount of stress-/hormone-responsive *cis*-acting elements were identified in their promoter regions, indicating they may play key roles in the regulation of development and stress responses of wild soybean. The GsSR proteins are predicted to be highly phosphorylated, and there are numerous interactions among GsSR proteins and other proteins. These GsSR members show different associations with spliceosome-associated components such as U1-70K or U2AF^35^, implying they may play different roles in pre-mRNA splicing process. Most *GsSRs* are highly expressed in stem and root tissues and are involved in salt and alkali stresses, and their splicing patterns were also in a tissue- and stress-dependent manner. In summary, these results will lay a foundation for us to explore the biological functions of SR proteins in plant growth, development, and stress responses.

## Figures and Tables

**Figure 1 ijms-25-11175-f001:**
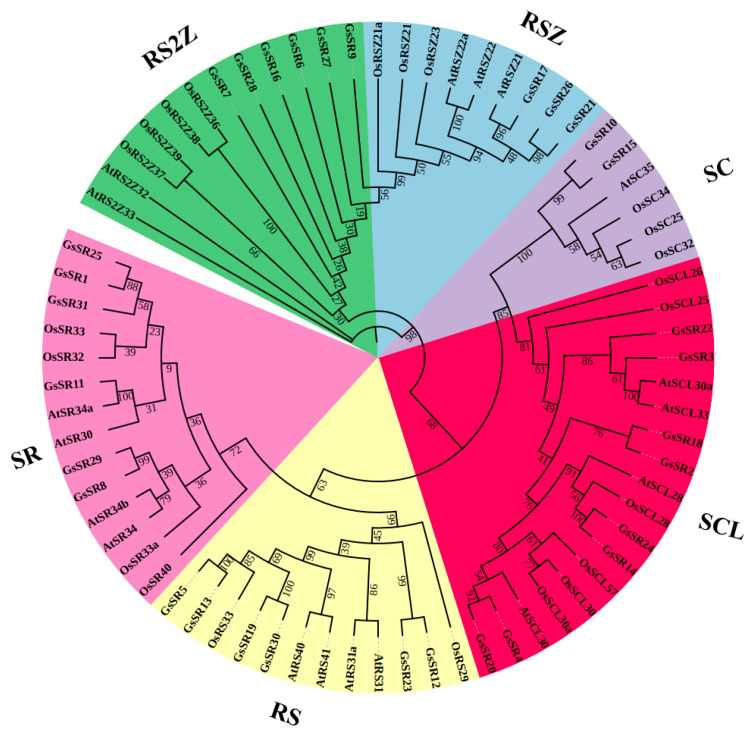
Comparative phylogenetic tree of GsSR, AtSR, and OsSR proteins. The maximum likelihood (ML) tree was constructed based on the amino acid sequences of SR proteins from *Glycine soja*, *Arabidopsis thaliana,* and *Oryza sativa* using IQ-tree incorporated in TBtools with 5000 bootstrap replicates. The different colors indicate different sub-families.

**Figure 2 ijms-25-11175-f002:**
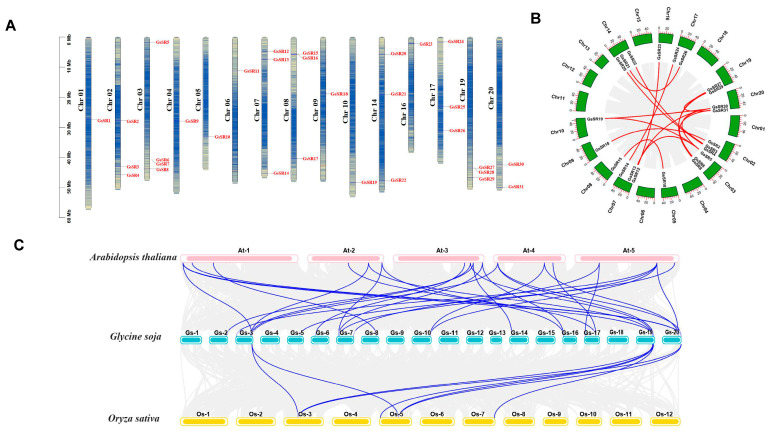
Chromosomal distribution and collinearity analysis of GsSR protein genes. (**A**) Chromosomal distribution of GsSR protein genes. The left scale determines the position of each *GsSR* on chromosome. Different shades of color reflect the distribution of gene density on chromosomes. (**B**) Inter-chromosomal relationships and segmental duplication of *GsSRs*. The green blocks indicate the part of wild soybean chromosomes (Chr01–Chr20). The duplicated GsSR protein gene pairs are highlighted in red lines. (**C**) Synteny analyses of SR protein genes between wild soybean and model plant species (Arabidopsis and rice). The gray lines indicate collinear blocks, and syntenic SR protein gene pairs are highlighted in blue lines.

**Figure 3 ijms-25-11175-f003:**
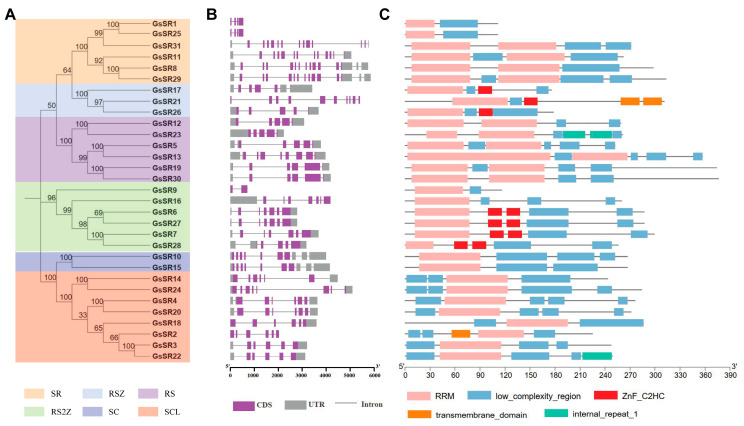
The gene structure and conserved protein motifs of 31 GsSR protein members. (**A**) The neighbor joining (NJ) tree was constructed using MEGA7 with 1000 bootstrap replicates. (**B**) Gene structure of GsSR protein genes. Grey boxes denote UTRs (untranslated regions); purple boxes denote exons; black lines denote introns. (**C**) Conserved motif analysis of GsSR proteins. A scale bar is provided at the bottom, and the length of each gene/protein is shown proportionally.

**Figure 4 ijms-25-11175-f004:**
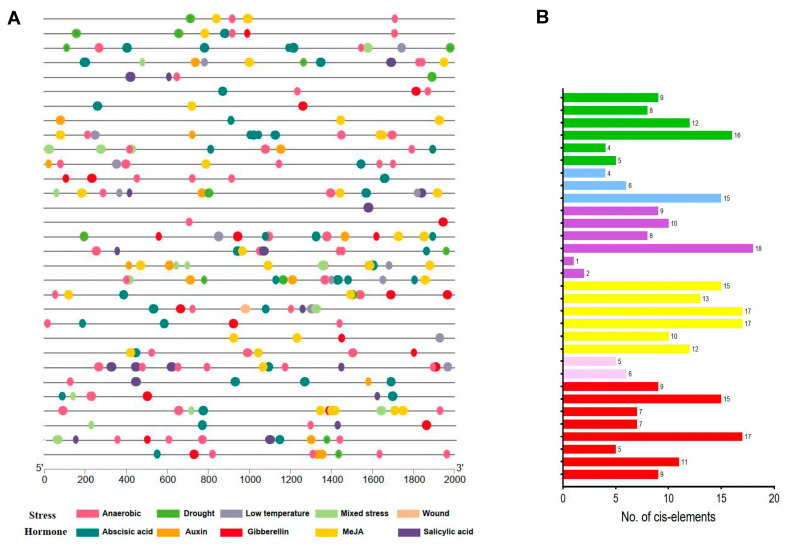
Search of stress/hormone-associated *cis*-acting elements on *GsSR* gene promoters. (**A**) Putative stress/hormone-associated *cis*-acting elements in the promoter regions of *GsSR* genes. The putative *cis*-acting elements were searched from the 2000 bp promoter regions in the upstream of coding sequences of *GsSR* genes by using PlantCARE database. (**B**) Statistics of putative stress/hormone-associated *cis*-acting elements.

**Figure 5 ijms-25-11175-f005:**
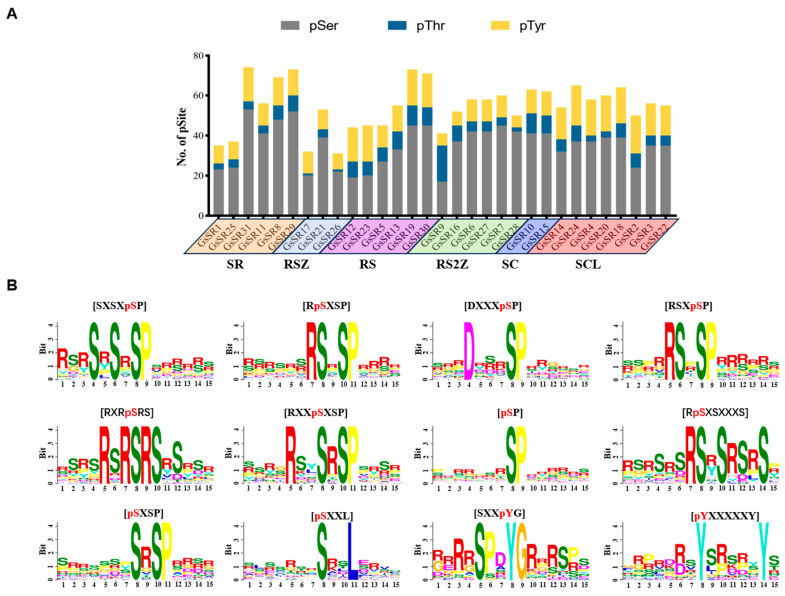
Prediction of phosphorylation sites on GsSR proteins. (**A**) The distribution of phosphorylated serine (pSer), phosphorylated threonine (pThr), and phosphorylated tyrosine (pTyr) sites on GsSR proteins; (**B**) the sequence logos of the conserved motifs around the phosphosites (pSer or pTyr) extracted from GsSR proteins.

**Figure 6 ijms-25-11175-f006:**
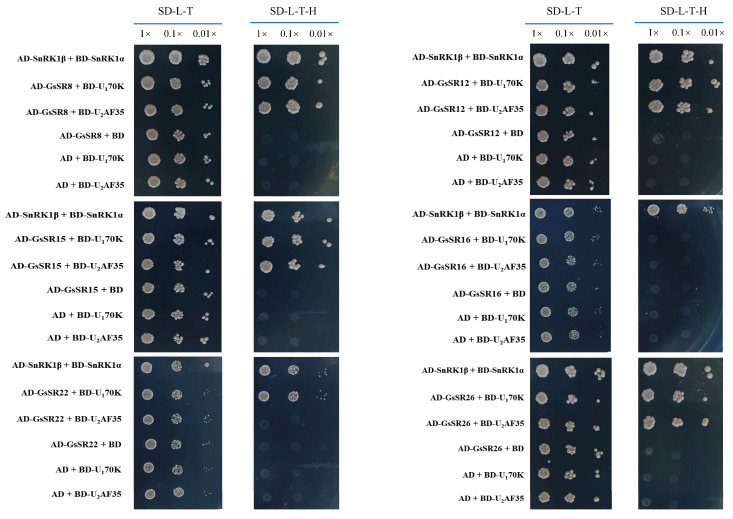
Physical interactions between representative GsSR proteins and snRNPs revealed by Y2H assay. Yeast cells carrying the indicated constructs were diluted and grown on SD/-Trp-Leu or SD/-Trp-Leu-His medium for 6 days at 28 °C.

**Figure 7 ijms-25-11175-f007:**
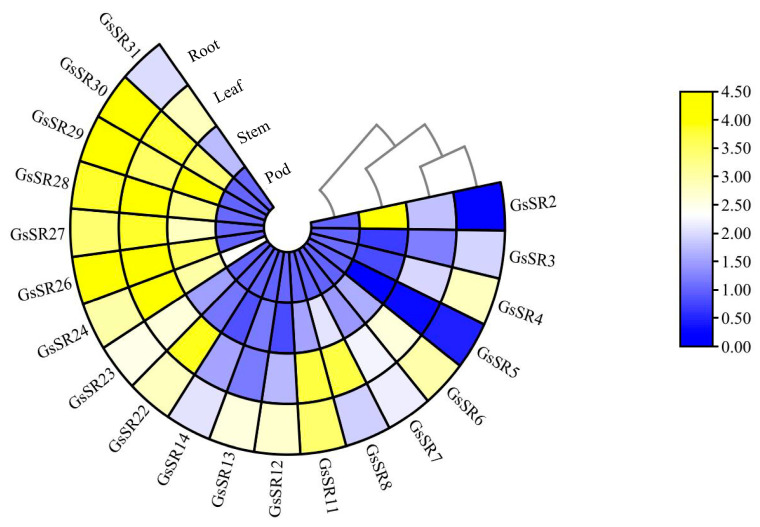
Heatmap representation of expression profiles of the selected 20 *GsSR* genes in different tissues of wild soybean.

**Figure 8 ijms-25-11175-f008:**
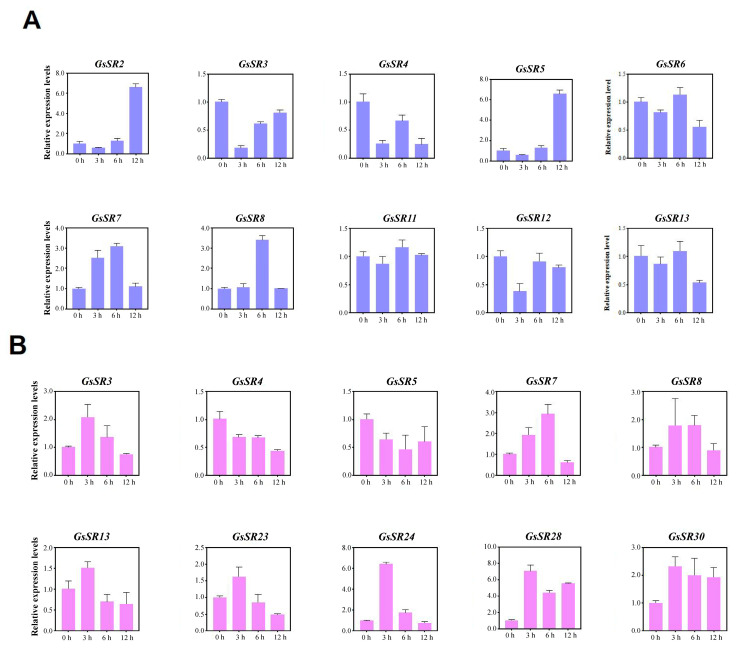
Expression patterns of *GsSR* genes from wild soybean under salt and alkali stresses. (**A**) Expression patterns of selected 20 *GsSR* genes under salt treatment (150 mM NaCl) for 0, 3, 6 and 12 h respectively; (**B**) expression patterns of selected 10 *GsSR* genes under alkali treatment (150 mM NaHCO_3_) for 0, 3, 6, and 12 h, respectively. The error bars represent the standard error of the means of three replicates.

**Figure 9 ijms-25-11175-f009:**
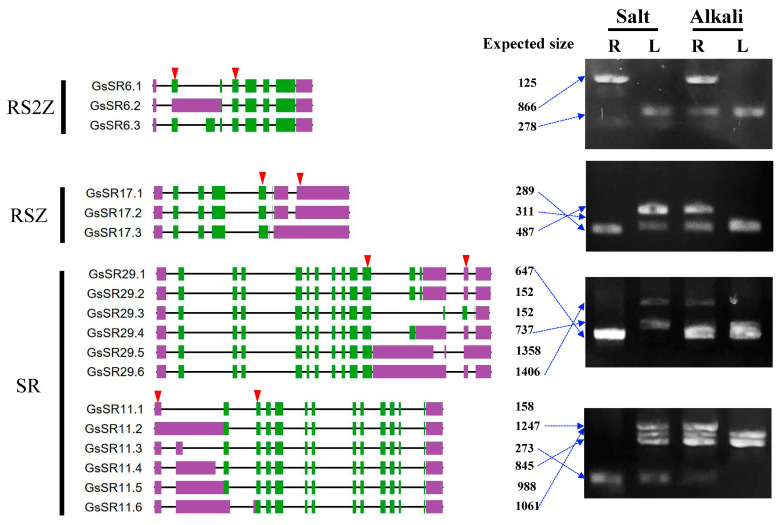
Analyses of splicing patterns of selected eight *GsSR* genes based on RT-PCR. The diagrams on the left are schematic diagrams of predicted alternatively spliced transcripts of *GsSR6*, *GsSR11*, *Gs17*, and *GsSR29*. Green boxes represent exons, purple boxes represent untranslated regions (UTRs), and black lines represent introns. The red arrowheads indicate the position of primers used for RT-PCR. The numbers in the middle indicate the expected sizes of PCR products. The diagrams on the right are the representative gel images of RT-PCR results. The wild soybean seedlings were treated with salt (200 mM NaCl) or alkali (100 mM NaHCO_3_) for 6 h. R: root, L: leaf.

**Table 1 ijms-25-11175-t001:** Basic information of GsSR protein family members.

Gene Name	*G.soja* ID	Sub- Family	Gene Location	cDNA(bp)	CDS (bp)	Peptide (aa)	pI	Molecular Weight (kDa)	Exons	Subcellular Location Prediction	No. of Phosphorylation Sites
GsSR1	Glysoja.01G000954	SR	Chr01: 27295247–27295784 (−)	336	336	111	10.11	13.25	4	Plastid	35
GsSR2	Glysoja.02G004178	SCL	Chr02: 27637432–27639458 (+)	678	678	225	10.05	26.99	6	Golgi	50
GsSR3	Glysoja.02G004721	SCL	Chr02: 43038628–43041828 (+)	1262	744	247	11.37	28.95	6	Nucl	56
GsSR4	Glysoja.02G004993	SCL	Chr02: 45706002–45709626 (+)	1302	831	276	11.22	31.09	8	Nucl	58
GsSR5	Glysoja.03G005795	RS	Chr03: 1689550–1693320 (−)	1299	759	252	10.16	29.49	6	Nucl	45
GsSR6	Glysoja.03G007465	RS2Z	Chr03: 40740780–40743564 (+)	1220	864	287	10.32	32.97	7	Nucl	58
GsSR7	Glysoja.03G007466	RS2Z	Chr03: 40745267–40748941 (+)	1338	900	299	10.17	34.04	7	Nucl	60
GsSR8	Glysoja.03G007832	SR	Chr03: 43760499–43766241 (−)	1865	897	298	10.32	33.73	15	Nucl	69
GsSR9	Glysoja.04G009752	RS2Z	Chr04: 27916825–27917542 (−)	351	351	116	6.90	12.96	2	Nucl	41
GsSR10	Glysoja.05G012047	SC	Chr05: 32949924–32953915 (+)	1922	804	267	11.39	31.26	10	Nucl	63
GsSR11	Glysoja.06G014615	SR	Chr06: 11212673–11217716 (−)	1217	789	262	10.39	29.21	13	Nucl	56
GsSR12	Glysoja.07G016978	RS	Chr07: 4715649–4718726 (+)	1487	780	259	9.83	30.58	5	Nucl	44
GsSR13	Glysoja.07G017217	RS	Chr07: 7282658–7286624 (−)	1934	1074	357	9.60	41.54	9	Nucl	55
GsSR14	Glysoja.07G018965	SCL	Chr07: 45207153–45211627 (−)	1174	732	243	11.05	28.35	7	Nucl	54
GsSR15	Glysoja.08G019871	SC	Chr08: 5603963–5608110 (+)	1909	804	267	11.37	31.04	10	Nucl	62
GsSR16	Glysoja.08G019889	RS2Z	Chr08: 5732435–5736603 (+)	1921	783	260	9.19	29.02	8	Nucl	52
GsSR17	Glysoja.08G021990	RSZ	Chr08: 40341249–40344664 (−)	1856	531	176	10.78	20.21	7	Nucl	32
GsSR18	Glysoja.09G023772	SCL	Chr09: 18881462–18885049 (+)	1292	861	286	10.58	33.12	6	Nucl	64
GsSR19	Glysoja.10G027976	RS	Chr10: 48349873–48353995 (−)	1486	1125	374	10.12	43.02	6	Nucl	73
GsSR20	Glysoja.14G037920	SCL	Chr14: 5735368–5739003 (−)	1296	816	271	11.05	30.86	8	Nucl	60
GsSR21	Glysoja.14G038546	RSZ	Chr14: 19065077–19070488 (−)	933	933	311	9.74	34.62	10	Golgi	53
GsSR22	Glysoja.14G039188	SCL	Chr14: 47787655–47790773 (+)	1291	750	249	11.42	28.97	6	Nucl	55
GsSR23	Glysoja.16G042491	RS	Chr16: 2081413–2083644 (+)	1807	786	261	10.04	30.84	5	Extracel	45
GsSR24	Glysoja.17G044660	SCL	Chr17: 1577936–1583020 (+)	1207	855	284	10.75	33.29	8	Plastid	65
GsSR25	Glysoja.17G046255	SR	Chr17: 23006391–23006928 (−)	336	336	111	10.11	13.20	4	Nucl	37
GsSR26	Glysoja.17G046405	RSZ	Chr17: 30974142–30977820 (+)	1192	537	178	11.41	20.41	5	Nucl	31
GsSR27	Glysoja.19G051819	RS2Z	Chr19: 43517530–43520317 (+)	1208	864	287	10.10	32.80	7	Nucl	58
GsSR28	Glysoja.19G051820	RS2Z	Chr19: 43522094–43525264 (+)	1514	771	256	9.86	28.62	6	Nucl	50
GsSR29	Glysoja.19G052171	SR	Chr19: 46520201–46526052 (−)	1848	942	313	10.26	35.27	15	Nucl	73
GsSR30	Glysoja.20G054252	RS	Chr20: 42489207–42493393 (+)	1525	1131	376	10.12	43.27	6	Nucl	71
GsSR31	Glysoja.20G055028	SR	Chr20: 49962213–49967989 (+)	930	816	271	10.21	30.98	14	Nucl	74

## Data Availability

The data that support the findings of this study are available from the corresponding authors upon reasonable request.

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
