# Peer review of "Genome-Scale Identification of Wild Soybean Serine/Arginine-Rich Protein Family Genes and Their Responses to Abiotic Stresses"

_ijms, 2024, doi:10.3390/ijms252011175_

Round 1

Reviewer 1 Report

Comments and Suggestions for Authors

The manuscript entitled "Genome-scale identification of wild soybean serine/arginine-rich protein family genes and their responses to abiotic stresses" provides a comprehensive study on the identification and characterization of serine/arginine-rich (SR) protein family genes in wild soybean (Glycine soja), with focusing on their role in abiotic stress responses. The study deploys various bioinformatics tools and experimental methods to present insights into the phylogeny, chromosomal distribution, gene duplication, protein interactions, and expression patterns of SR proteins under stress conditions. which contributes to understanding the molecular mechanisms underlying stress adaptation in legumes and is a critical area for improving crop resilience. The manuscript makes a valuable contribution to the field of plant molecular biology by elucidating the role of SR proteins in wild soybean's stress responses and is well-suited for publication in the International Journal of Molecular Sciences.

However, the manuscript could benefit from improvements in clarity and focus. While the study covers many aspects of GsSR proteins, a clearer focus on specific hypotheses or research questions would enhance readability. A more structured presentation of results and discussions is recommended. The discussion section could be expanded to compare findings with existing literature more thoroughly, highlighting novel contributions and potential implications for crop improvement. Some minor grammatical errors and problematic phrasings should be corrected to improve clarity. For example, "the exon-intron structure is relatively conserved within a sub-family" can be rephrased for grammatical consistency.

Comments on the Quality of English Language

Minor editing of English language required.

Author Response

The manuscript entitled "Genome-scale identification of wild soybean serine/arginine-rich protein family genes and their responses to abiotic stresses" provides a comprehensive study on the identification and characterization of serine/arginine-rich (SR) protein family genes in wild soybean (Glycine soja), with focusing on their role in abiotic stress responses. The study deploys various bioinformatics tools and experimental methods to present insights into the phylogeny, chromosomal distribution, gene duplication, protein interactions, and expression patterns of SR proteins under stress conditions. which contributes to understanding the molecular mechanisms underlying stress adaptation in legumes and is a critical area for improving crop resilience. The manuscript makes a valuable contribution to the field of plant molecular biology by elucidating the role of SR proteins in wild soybean's stress responses and is well-suited for publication in the International Journal of Molecular Sciences.

Thank you very much for your positive comments

However, the manuscript could benefit from improvements in clarity and focus. While the study covers many aspects of GsSR proteins, a clearer focus on specific hypotheses or research questions would enhance readability.

Thank you very much for pointing out this issue. We have accordingly re-organized the manuscript according to your comments and advice.

A more structured presentation of results and discussions is recommended. The discussion section could be expanded to compare findings with existing literature more thoroughly, highlighting novel contributions and potential implications for crop improvement.

We have expanded the discussion, and removed and added some literatures.

Some minor grammatical errors and problematic phrasings should be corrected to improve clarity. For example, "the exon-intron structure is relatively conserved within a sub-family" can be rephrased for grammatical consistency. Minor editing of English language required.

We have systematically corrected and revised the manuscript. The revised portions in the new version are highlighted in green.

Reviewer 2 Report

Comments and Suggestions for Authors

The authors seem to have analyzed three-week old soybean plants after up to 12 hours of salt or base treatment. They do not state at what stage they sampled the plants, nor do they say how many plants were individually sampled and analyzed. Three-week old soybean plants do not have very much leaf material, and have no pods. The authors must describe more fully their experimental setup, including statistical sampling.

In the revised version, labels on the bar graphs must be enlarged.

Author Response

The authors seem to have analyzed three-week old soybean plants after up to 12 hours of salt or base treatment. They do not state at what stage they sampled the plants, nor do they say how many plants were individually sampled and analyzed. Three-week old soybean plants do not have very much leaf material, and have no pods. The authors must describe more fully their experimental setup, including statistical sampling.

Thank you very much for you pointing out these problems. We have re-worded this part showing in line 471-475. The samples for spatial expression (different tissues) and temporal expression (stress treatments) are from wild soybean plants with different ages.

In the revised version, labels on the bar graphs must be enlarged.

The labels on the bar graphs of figures 8 have been enlarged.

Round 2

Reviewer 2 Report

Comments and Suggestions for Authors

Comparing the genome of wild soybean to that of Arabidopsis is standard, since everything from slime mold to maple trees are compared to that. Comparing wild soybean to rice, and then pointing out that there is not much overlap, is like comparing rocks to trees-- both are found attached to the ground and both can be used to build houses, but so what? Rice is a monocot, and soy is a dicot (like Arabidopsis, which is from a very different genus, and which, like rice, cannot fix nitrogen) and it is not surprising that the two crops are not that similar. It would have been much more interesting to compare SR regions of wild soybean to those of domestic soybean, and to see where there are relatively simple opportunities for breeding or otherwise introducing stress resistance into the cultivated crop. Is there a good reason why the authors did not follow this line of research? If so, they should put their reasoning into a new version of the manuscript.

Author Response

(The authors gave the same response as above.)

Round 3

Reviewer 2 Report

Comments and Suggestions for Authors

Authors did not respond to my question as to why they did not compare GsSR regions with GmSR regions. Instead, they sent the response from version 2 of their manuscript.

Author Response

1. Authors did not respond to my question as to why they did not compare GsSR regions with GmSR regions. Instead, they sent the response from version 2 of their manuscript.

Thank you very much for your pointing out this problem. Actually, we are doing this work (comparison of SR genes and proteins between the wild and cultivated soybeans) as a separate project. Phylogenetically, the SR genes and proteins share high similarities between these two species. However, we indeed identified many SNPs and indels in the gene coding and regulatory sequences. As what you indicated, we are trying to identify the SR genes which may play important roles in salt-resistance of wild soybean and apply the candidate genes in cultivated soybean breeding. We added a few sentences in the Discussion portion (line 298-301).
